# Riesite, a New High Pressure Polymorph of TiO$_2$ from the Ries Impact Structure

**Oliver Tschauner [1],\*, Chi Ma [2] , Antonio Lanzirotti [3] and Matthew G. Newville [3]**

[1]  Department of Geoscience, University of Nevada Las Vegas, Las Vegas, NV 89154, USA
[2]  Division of Geology and Planetary Sciences, California Institute of Technology, CA 91125, USA; chi@gps.caltech.edu
[3]  GeoSoilEnviroCARS, University of Chicago at the Advanced Photon Source, Argonne National Laboratory, Argonne, IL 60367, USA; lanzirotti@uchicago.edu (A.L.); Newville@uchicago.edu (M.G.N.)
\*  Correspondence: oliver.tschauner@unlv.edu

**Abstract:** This paper describes riesite, a new high-pressure polymorph of TiO$_2$ from the Ries impact structure, Germany. Riesite occurs in shock-induced melt veins within xenoliths of bedrock in suevite. It is structurally closely related to srilankite from which it differs by having two distinct cation sites rather than one and through its monoclinic symmetry. It is indicative that riesite forms only upon release from the shock state upon back transformation from akaogiite.

**Keywords:** high-pressure mineral; high pressure; mineral physics; impacts

## 1. Introduction

Rutile is a common accessory mineral in many rocks of the crust. At low pressure, its stability field is constrained by transformation into anatase at high temperature and to brookite at intermediate pressures and temperatures. At high pressure, rutile undergoes a sequence of transitions to phases isotypic with scrutinyite ($\alpha$-PbO$_2$) [1–3], baddeleyite [4,5], and cottunite (PbCl$_2$O) [6,7]. Baddeleyite-type TiO$_2$ is an approved mineral with the name akaogiite [8]. The boundaries between these polymorphs depend on temperature but also on composition; the solubility of Zr in rutile increases with increasing temperature [9], but the boundary between rutile and scrutinyite-type TiO$_2$ is shifted to lower pressures with increasing Zr-content [10]. Nb and Ta exert a similar effect, but the structural relations between rutile- and post-rutile phases are more complex than for Zr because the incorporation of Nb and Ta in rutile requires coupled substitution, which results in cation ordering under most conditions. Ixiolite is scrutinyite-type—Fe-Ti-Nb-Ta-oxide [11]. Naturally occurring scrutinyite-type TiO$_2$ is also an approved mineral with the name srilankite [12]. Most srilankites in terrestrial rocks contain about 40–60 mol % ZrO$_2$ [12–14]. Endmember or near to endmember srilankite has been reported from impact sites [15,16] and from ultrahigh pressure metamorphic rocks [17]. Unfortunately, srilankite from those occurrences has been labeled as "TiO$_2$-II" as in the earlier experimental studies on high-pressure polymorphs of TiO$_2$ [1,4,6], while it should properly be labeled as srilankite, a mineral known since 1983 [12]. The fact that srilankite was originally reported with chemical formula ZrTi$_2$O$_6$ but with scrutinyite-structure corresponding to a structure formula (Ti,Zr)O$_2$ [12] has certainly contributed to the confusion. Natural occurrences, as well as the majority of experimental studies, indicate a sequence of pressure-induced transformations: rutile → srilankite → akaogiite with transition pressures 10–15 and 20–25 GPa at 300 K, respectively (2,3,7). This range of phase transitions and the abundance of TiO$_2$ as accessory phases in the continental crust makes high-pressure polymorphs of TiO$_2$ good pressure markers for impact events [14,15,18]. This includes studies of the evolution of particular impact sites [14,15] as well as the assessment of the frequency of impacts in the ancient geologic past

because rutile is a common detrital mineral and its high-pressure polymorphs may be conserved in detrital grains as well, similar to reidite that has been found in detritral zircon grains [19].

Here we describe structure, composition, and occurrence of a new high-pressure polymorph of $TiO_2$, riesite. Riesite has been approved as a new mineral species with number IMA-2015-110 [20].

## 2. Materials and Methods

Riesite was found in a thin section (ZLN114c) from a xenolith of garnet-sillimanite restite with shock-melt veins that was trapped in suevite. This xenolith and the mineralogy of the shock-melt vein have been described in an earlier study [21], to impact-related melt veins which carry high-pressure minerals whose formation requires several GPa pressure or more. This is in distinction to impact-related pseudotachylites which do not contain such high-pressure minerals. The high-pressure polymorphs of common accessories like rutile and zircon permit distinction between both types of shock-induced melt veins. In the current case, the melt vein contains majorite-rich garnet [21], a jadeite-rich clinopyroxene, and accessory akaogiite, and reidite (the high-pressure polymorph of zircon). Akaogiite and reidite had previously been reported from the Ries [8,22,23]. Entrapped garnet clasts contain stishovite, which probably formed through direct solid state transformation of quartz inclusions [21]. Since an extensive discussion of the petrography and mineralogy of the xenolith and on the composition of the shock melt vein has been presented by Stähle et al. [21] we will not recapitulate these findings here and instead will focus on the observation and characterization of riesite. Within the shock melt vein, we observed clasts of $TiO_2$ and $FeTiO_3$. These clasts are opaque in the optical microscope but yield a high back scatter electron ('BSE') intensity region in the scanning electron microscope ('SEM') (Figure 1a,b). At higher magnification, we find that the titanium dioxide clasts are composed of grains of sub-micrometer in diameter (Figure 1c).

Electron backscatter diffraction ('EBSD') reveals that these grains are not from a phase with rutile structure, although the composition is close to pure $TiO_2$ (Table 1). In one of the cases examined, a fine grained dense aggregate of this phase encloses a highly deformed rutile kernel (see below). Quantitative elemental microanalyses of these clasts were carried out at Caltech using a JEOL 8200 electron microprobe operated at 15 kV and 20 nA in focused beam mode. Analyses were processed with the CITZAF correction procedure [24]. Ten point analyses of type riesite were averaged and the results are given in Table 1.

**Table 1.** Analytical data for riesite. Average of ten point analyses. The empirical formula (based on 2 O atoms pfu) is $(Ti_{0.997}Fe_{0.005})O_2$.

| Constituent | Wt % | Range | SD | Probe Standard |
|---|---|---|---|---|
| $TiO_2$ | 99.25 | 98.98–99.62 | 0.19 | $TiO_2$ |
| FeO | 0.42 | 0.33–0.53 | 0.07 | Fayalite |
| CaO | 0.03 | 0.02–0.04 | 0.01 | Anorthite |
| Total | 99.70 | - | - | - |

We collected X-ray microdiffraction data at the undulator beamline 13-IDE (GSECARS, APS, Argonne National Laboratory) using a primary beam of wavelength at 0.6199 Å, monochromatized by a two-crystal Si monochromator. The X-ray beam was focused to $2 \times 3$ μm$^2$ by vertical and horizontal Kirkpatrick-Baez mirrors of 200 mm focal length. A MAR165 CCD area detector was used for collecting diffraction data in forward scattering geometry. The thin section that contains riesite was initially examined by X-ray fluorescence mapping with same spatial resolution of $2 \times 3$ μm$^2$ in order to identify Ti-rich clasts in the shock-melt vein of section ZLN114c. Clasts with rutile composition were then examined by X-ray diffraction mapping in 2 μm steps horizontally and vertically through the focused X-ray beam over the selected region of the clasts and diffraction patterns recorded at each step. The diffraction pattern images were corrected for diffuse scattering from the glass slide of the thin section using background image subtraction in Fit2D [25]. Then the patterns were corrected for

geometric distortion from detector tilt using the GSE-ADA analysis software [26] and integrated using Fit2D [25]. Riesite was observed together with ilmenite, rutile, and akaogiite in several transformed or partially transformed rutile clasts. Initially we interpreted riesite as Ti-srilankite, but soon we found that deviations of observed peak intensities and their 2ϑ angles from the modelled pattern of TiO$_2$-srilankite reflect an actual reduction in crystal structure symmetry (Figure 2a,b). The possible sub-group related structures were examined. The more promising model structures were used for LeBail extraction of structure factor moduli and examined by simulated annealing [27,28] without symmetry bias (in space group *P1*). After convergence, automatic symmetry search resulted consistently in a monoclinic distortion of the srilankite structure into subgroup *P2/c* in setting *P1 2/c 1*.

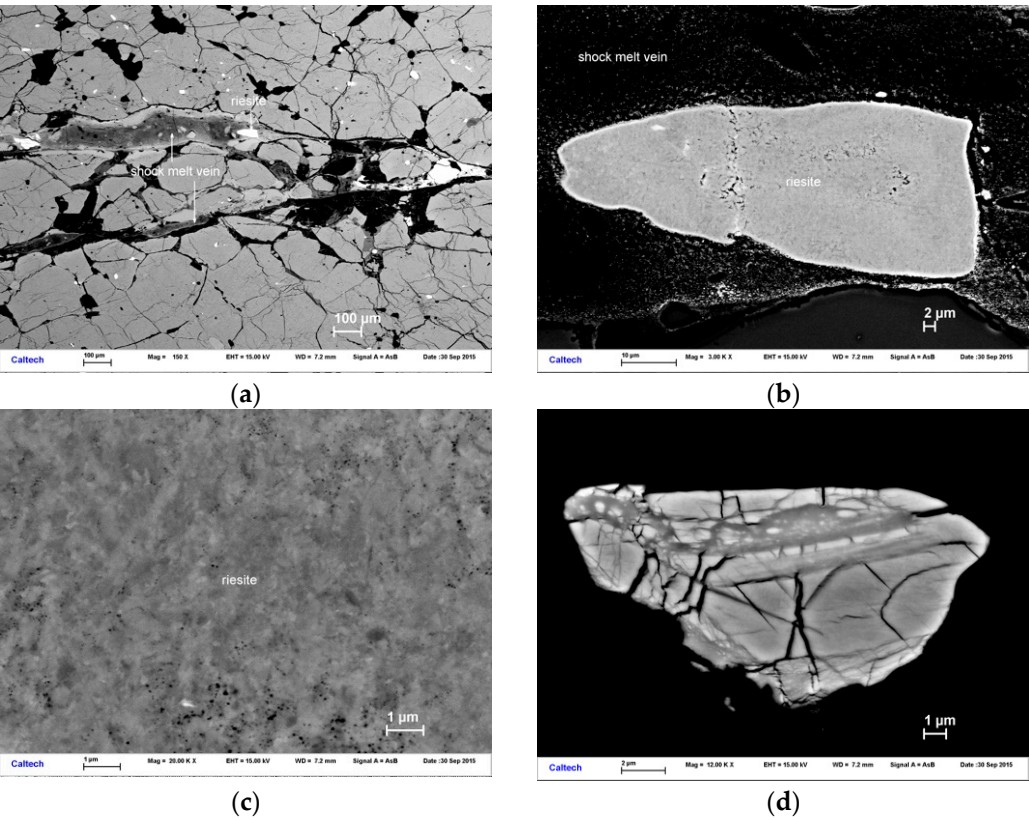

**Figure 1.** Field Emission secondary electron microscope (SEM) back scatter electron (BSE) images of riesite type material at different magnification: (**a**) Overview image showing the location of a riesite clast within the shock melt vein which cuts through the ZNL114c thin section. (**b**) The same clast of type riesite at higher magnification. Riesite is pseudomorph after rutile, replacing a former single crystal through a fine-grained aggregate. The enhanced brightness at the border between the riesite aggregate and the shock melt vein is result of electrostatic charging. (**c**) Area of the same clast of riesite, shown at high magnification. A bright crystallite of submicrometer dimensions is Zr-rich. Type riesite contains no significant Zr. (**d**) Reidite lamellae in a zircon clast trapped in the shock melt vein of ZLN114c. Reidite was identified by electron backscatter diffraction (EBSD).

For powder data in many cases (and in this particular case), indexation cannot be well conducted independent from structure modeling. Thus, we did not rely on forward modeling of monoclinic cells but used the full set of integrated intensities of the new TiO$_2$-phase for reversed Monte Carlo (rMC) modeling (local optimization) without symmetry bias (that is: in space group *P1*). rMC converged to an F-based refinement factor $R_F$ of less than 7% and subsequent automatic symmetry search gave consistently the riesite cell and space group (*P2/c*, SG number 13) for different sets of intensities obtained through LeBail extractions from diffraction patterns from different locations and paragenesis: riesite and rutile (Figure 2a,b), as well as patterns with riesite, ilmenite, and akaogiite (Figure 2c). Thus, the

choice of space group and cell dimensions is robust and is based on a large statistical set of data obtained from several patterns of riesite-bearing sample material. We point out that the srilankite-model gives an $R_F$ worse by factor of two than the monoclinic structure. EBSD patterns match better to the monoclinic structure with a mean angular deviation of 0.38° than to the orthorhombic structure with a mean angular deviation of 0.47° (Figure 2d). The main difference between both structures is in the site symmetry and fractional atomic coordinates of Ti. Since EBSD does not directly probe these parameters, it is not very sensitive to their differences.

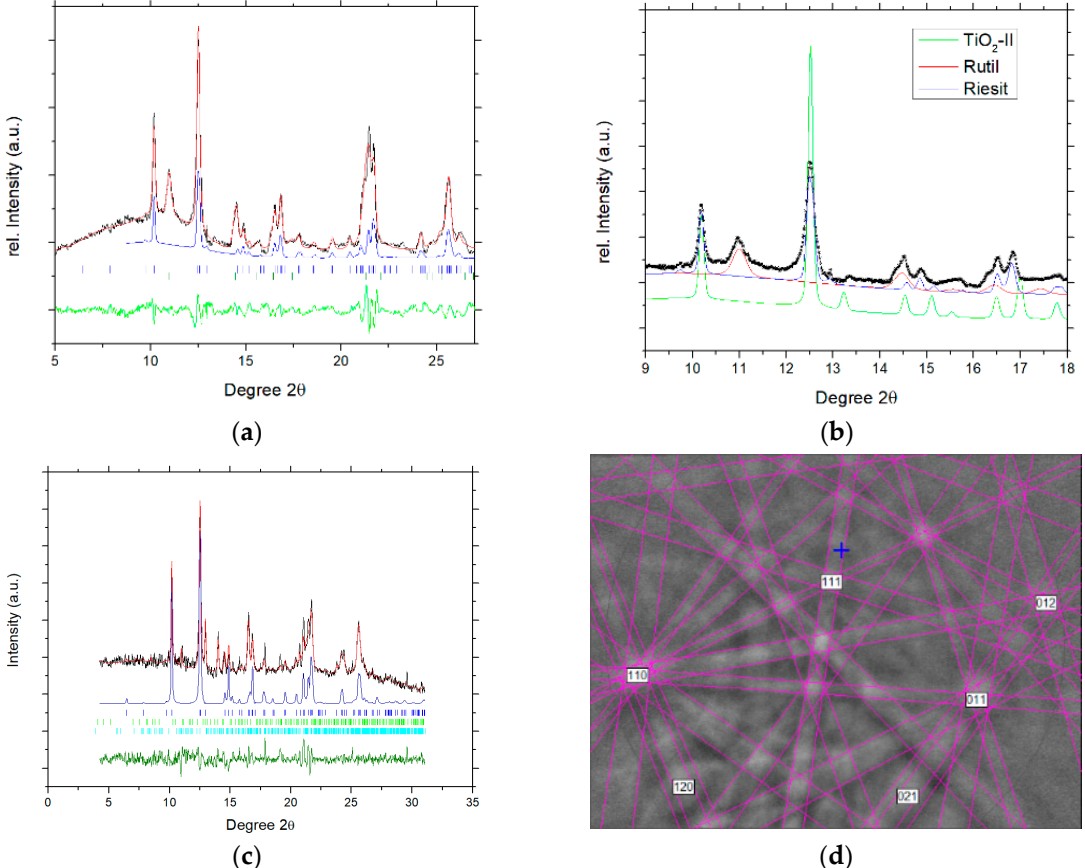

(a)

(b)

(c)

(d)

**Figure 2.** X-ray diffraction and EBSD data of riesite: (**a**) Representative X-ray diffraction pattern of type riesite. Black crosses = observed data, red line: Rietveld refined pattern of riesite plus rutile. The pattern represents ~70 mass% riesite and 30 mass% rutile. Blue line: Contribution of riesite to the Rietveld-refined pattern. (**b**) The same pattern between 9 and 19° 2ϑ with the calculated pattern of Ti-endmember srilankite added (green line). The same profile parameters and scale factor as for riesite were used. Ti-Srilankite clearly mismatches the observed pattern. (**c**) Rietveld refinement of diffraction pattern from a different clast of riesite, coexisting with ilmenite and akaogiite. (**d**) Observed and fitted EBSD pattern of type riesite. The mean angular deviation of the fit was 0.38°.

## 3. Results

We used this monoclinic distorted structure for structure refinement with the Rietveld method using Powdercell [29] and GSAS [30]. Pseudovoigt peak profiles were used with Gaussian terms $U = 728$, $V = 8.1$, $W = 7.3$ and Lorentzian terms $Lx = 6.3$ and $Ly = 8.1$ (in GSAS). Rietveld Refinement converged to a weighted refinement factor $R_{wp}$ of 6.8% (Figure 2a), whereas the orthorhombic Ti-srilankite structure converges to not better than 11.1% (Figure 2b). As a further refinement measure we obtained a profile-based refinement factor $R_p$ of 5.1% and $\chi^2 = 11.2$ for 1534 observations. $R_{F^2}$ was 9.6% and $R_F$ was found to be 7.8%. LeBail refinement of riesite converged to an $R_p$ of 5.0%. The pattern used for final structure refinement of riesite (Figure 2a) contained 71 mass% riesite and 29 mass% rutile.

An additional refinement of riesite from an aggregate with ilmenite and remnant rutile (Figure 2c) gave an $R_p$ of 5.7% and a $R_{wp}$ of 7.8%.

## 4. Discussion

We place the new mineral riesite into its context with other high-pressure polymorphs of rutile. Riesite assumes space group *P2/c*, which is a direct subgroup of the space group of srilankite, *Pnma*. However, in riesite the cations reside on different Wykhoff sites than in srilankite. A direct group-subgroup relation maps the cations from srilankite onto Wyckoff-sites 2e and 2f in the monoclinic structure. However, our structure analysis converged persistently to a structure with the cations on site 4g rather than 2e and 2f. In other words, the present phase is not a product of a simple distortive lattice relaxation of Ti-rich srilankite into a monoclinic structure but reflects a significant cation sublattice shift relative to the direct subgroup structure of srilankite of between 5% and 22% (Ti1 coordinate y/b):

$$Ti2: \Delta(x|y|z) = 0.022|0.018|0.041$$

$$Ti1: \Delta(x|y|z) = 0.038|0.040|0.015$$

Thus, despite the small deviation of the beta angle from 90 degrees, the change in axes length generates overall splitting of powder diffraction lines and the difference in cation lattice causes changes in the observed intensities that make the powder patterns of riesite distinct from srilankite. Figure 2b shows clearly that the observed pattern cannot be matched by the orthorhombic srilankite cell and structure.

This fundamental difference between riesite and srilankite brings us to the formation mechanism of riesite. Riesite assumes a space group that is a direct subgroup of the space group of srilankite. At first glance this suggests riesite to form upon a distortive transition that is guided by this group-subgroup relation as one of the Landau-criteria. As we already showed above, this is not the case: The continuous transition from direct group-subgroup transformation *Pnma* → *P2/c* with index 2 gives the following mapping of Wyckoff sites:

$$Ti: 4c \rightarrow 2e + 2f$$

$$O: 8d \rightarrow 4g + 4g$$

However, in riesite the cations clearly reside on sites 4g with half occupancy (Table 2). Internal cation shift between riesite and monoclinic distorted srilankite is marked with changes of atomic position between 5% (mostly) and 22% (Ti1 coordinate y/b). In other words, there is no direct group-subgroup relation that maps the cation sublattice of srilankite onto that of riesite.

**Table 2.** Atom coordinates of riesite. The structure is monoclinic and assumes space group number. 13, *P12/c1* with unit cell dimensions $a$ = 4.519(3) Å, $b$ = 5.503(8) Å, $c$ = 4.888(2) Å, $\beta$ = 90.59(8)°, $V$ = 121.5(1) Å$^3$. Site fractional occupancies (SFO) and isothermal displacement factors $B_{iso}$ are given. For comparison, the unit cell of endmember srilankite (TiO$_2$-II) is 4.5318(7), 5.5019(7), 4.9063(6) Å, respectively, with all angles 90°, and the fractional atomic coordinates are 0, 0.1704(3), $\frac{1}{4}$ (Ti) and 0.2716(6), 0.3814(7), 0.4142(7) (O) [31].

| Atom | Wyckoff | x/a | y/b | z/c | SFO | $B_{iso}$ (Å$^2$) |
|---|---|---|---|---|---|---|
| Ti1 | 4g | 0.041(2) | 0.142(4) | 0.268(2) | 0.47(3) | 0.6(1) |
| Ti2 | 4g | 0.51(1) | 0.311(7) | 0.78(1) | 0.53(1) | 0.7(1) |
| O1 | 4g | 0.28(1) | 0.36(1) | 0.436(7) | 1 [1] | 0.9(2) |
| O2 | 4g | 0.224(1) | 0.086(6) | 0.937(7) | 1 [1] | 0.9(2) |

[1] Fixed based on chemical analysis.

However, space group and Wyckoff site symmetries of the riesite structure can be obtained, through relation to akaogiite.

$$P1\ 2_1/c\ 1\text{(akaogiite)} \rightarrow P1\ c\ 1\text{(transient)} \rightarrow P1\ 2/c\ 1\text{(riesite)}$$

which maps all cations and anions onto sites 4g with partial occupancy. In riesite, there is no indication for partially occupied O-sites (as obtained through this mapping), but the cation sites are clearly partially occupied and of site symmetry 4g. We emphasize that riesite and akaogiite are distinct structures. Riesite is not a variety of akaogiite. The group-subgroup-chain *P1 2_1/c 1 → P1 c 1 → P1 2/c 1* involves a direct group-subgroup mapping from *P2_1/c* onto *Pc*. However, the mapping of *Pc* into *P2/c* is klassengleich not translationsgleich, whence it involves a sublattice-shift. Keeping the atomic fractional coordinates of akaogiite, this group-subgroup chain results in a fluorite-like arrangement, whereof in riesite half of the Ti-sublattice is shifted along [110] (in compliance with the klassengleiche mapping from *Pc* into *P2/c*).

Akaogiite is a polymorph of $TiO_2$ that occurs at higher pressure than srilankite. According to our group-theoretical analysis above, riesite assumes a structural state in between srilankite and akaogiite. One may expect riesite to form in a pressure-temperature regime intermediate between srilankite and akaogiite but it has not been reported from experiments. The fact that the riesite structure involves low-symmetric partially occupied cation sites is indicative for formation through a different process than static compression. The sublattice disorder and the structural relation between akaogiite and riesite suggest that riesite is the product of retrograde transformation of akaogiite upon rapid decompression and cooling. In this case, riesite does not have a thermodynamic stability field but forms at elevated temperature under similar pressures as srilankite in dependence of the path (rapid decompression of akaogiite at elevated temperature). In fact, the riesite type material has been found in exactly this environment: a shock melt vein in an impact-metamorphosed bedrock fragment. The density of riesite is $4.37 \pm 0.11$ g/cm$^3$ (Table 2), which is in between rutile (4.25 g/cm$^3$) and srilankite (4.38 g/cm$^3$ [12]), but below that of akaogiite at reference conditions (4.72 g/cm$^3$ [8]).

In general, the occurrence of riesite along with akaogiite is expected to be common in impact-metamorphism of rutile-bearing rock. In these environments, riesite indicates that peak pressures have been in the stability field of akaogiite above 20 to 25 GPa. Occurrence of Ti-endmember srilankite instead of riesite indicates that the peak pressures were below 20 GPa.

In the Ries xenolith ZLN114, riesite occurs as complete or partial replacement of rutile clasts trapped within a shock melt vein composed of majoritic garnet and jadeite. In partially transformed clasts, we find akaogiite, riesite, and very fine-grained rutile at the outer rim of the clasts, akaogiite further inside, and residual, highly strained rutile crystallites in the kernel of larger clasts. Reidite (Figure 1d) and stishovite are other high-pressure minerals identified in clasts adjacent to the type material of the new mineral.

In an earlier study of similar shock-melt veins in amphibolite xenoliths from Ries suevite, Stähle et al. [23] estimated peak shock pressure up to 17–20 GPa based on garnet barometry. However, the present paper is the first assessment of shock pressures for a xenolith of garnet-cordierite-sillimanite restite. The observation of riesite and akaogiite is consistent with peak pressure 20–25 GPa. The presence of reidite (Figure 1d) is also consistent with pressures above 12 GPa, if temperatures were above 1200 K [32]. Lower temperature shifts the transition from zircon to reidite to higher pressure [32]. We note that the shock-driven direct conversion along the Hugoniot of zircon and on experimental time scales of less than one ms occurs above 60 GPa [33]. Observation of liebermannite at the rim of the shock melt vein and majoritic garnet in the vein indicates that peak shock pressure was not above 20–22 GPa [34]. We note that this pressure range is markedly below the pressure in the isobaric core of the Ries impact event. Xenoliths like ZLN114 that were trapped in suevite record different peak-shock pressures and release paths depending on their original position relative to the isobaric core and their entrapment in suevite [35].

**Author Contributions:** O.T. collected X-ray diffraction and –fluorescence data, analyzed crystal structures, and contributed to the interpretation of data and writing of the manuscript, C.M. conducted chemical analysis, EBSD analysis, collected FE-SEM images, and contributed to the interpretation of data and writing of the manuscript, A.L. and M.G.N. set up experiment for X-ray diffraction and –fluorescence analysis, and contributed to the interpretation of data and writing of the manuscript. All authors have read and agree to the published version of the manuscript.

**Funding:** This work was supported by DOE Award DESC0005278, NSF EAR-1128799, DE-FG02-94ER14466, NSF Grants EAR-0318518, and DMR-0080065. Portions of this work were performed at GeoSoilEnviroCARS (The University of Chicago, Sector 13), Advanced Photon Source (APS), Argonne National Laboratory. GeoSoilEnviroCARS is supported by the National Science Foundation – Earth Sciences (EAR – 1634415) and Department of Energy- GeoSciences (DE-FG02-94ER14466). This research used resources of the Advanced Photon Source, a U.S. Department of Energy (DOE) Office of Science User Facility operated for the DOE Office of Science by Argonne National Laboratory under Contract No. DE-AC02-06CH11357.

**Acknowledgments:** We would like to thank Robert C. Liebermann for his suggestion to submit this paper to the Special Issue of *Minerals* in memory of Orson Anderson. We also thank V. Stähle and D. Stöffler for providing the thinsection ZLN114c which contains type riesite and we thank the Institut für Geowissenschaften, University of Heidelberg and in particular M. Trieloff for permitting us to examine this section, which is part of their collection. The publication fees for this article were supported by the UNLV University Libraries Open Article Fund.

**Conflicts of Interest:** The authors declare no conflict of interest. The funders had no role in the design of the study; in the collection, analyses, or interpretation of data; in the writing of the manuscript, or in the decision to publish the results.

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
