# Peer review of "Riesite, a New High Pressure Polymorph of TiO2 from the Ries Impact Structure"

_minerals, doi:10.3390/min10010078_

Round 1

Reviewer 1 Report

December 2019

Overview

This manuscript reports the discovery and documentation of a new TiO2 polymorph, Riesite. The mineral is an approved mineral by IMA, and is so named after the sample it was found in, which originated from the Ries impact structure in Germany. The authors present different types of diffraction data (X-ray microdiffraction, Rietveld refinement, EBSD), compositional data (EMPA), and some images (SEM-BSE) to document the crystallography of the discovered phase. In particular, emphasis in the Discussion is place on why this phase is not a variant of either srilankite or akaogiite. 

In general, I found the manuscript overall well-prepared, the data appear to be of high quality, and the conclusions soundly argued. Knowledge of this phase will be of interest to planetary scientists, those that study impact craters, metamorphic petrologists, and other communities.

The main issues I identified in this manuscript that could benefit from attention are relatively minor, but that I think would help the presentation. These include some minor text editing, removing some typos, and a bit more sample-related documentation. In particular, I would like to see the addition of a few figures, so long as their inclusion does not result in the manuscript exceeding the length limit set by the journal. All things considered, my suggestion is to publish after moderate revision.

Main issues:

(1) Can the authors include a TiO2 phase diagram with a suggestion for where riesite might reside? It is indicated in the text, but a phase diagram (even somewhat schematic) would be helpful.

(2) The authors claim that reidite occurs adjacent to the riesite, and attribute petrographic description of the sample to Stahle et al. However, these authors do not mention reidite. Riedite is an important phase that interests a lot of people- can the authors include an image of a riedite-bearing zircon that occurs in the same sample? It adds another element to constrain the P history, and does not appear to be documented anywhere else for this sample.

(3) The authors describe how riesite occurs together with ilmenite, rutile, and akaogiite- this sounds wildly interesting. Can the authors include an EBSD phase map to show this?

(4) The X-ray mapping is described as occurring over a 2x3 um2 area, that moved in 2 um steps. If the grain size of riesite is sub-micron, and also occurs as rims around ‘rutile kernals’, how was this dealt with given the much larger size of the X-ray beam? The authors should mention why this was/was not an issue in data collection/interpretation.

(5) Many abbreviations occur in the manuscript. Some are followed by references, but many are not, and I think should be defined. Among these are EBSD, SEM, Rf, rMC, Rwp, Rp, and others.

(6) In Table 2, can the authors please include data for srilankite as well, to facilitate comparison (it would also be nice to include data for akaogiite)

(7) In Figure 1, can the surrounding phases be identified? In panels A and B, can an inset box be included to show the locations of panels B and C? Can the authors include an EBSD map (band contrast, and/or phase map) of the grain shown in panel B? Its large enough that it should produce a lovely map, and would be the first of its kind. It would also inform about its state of strain.

(8) In the caption to Figure 2, switch the order of the first 4 words, as the first panels show X-ray images, and the second shows ebsd.

Specific comments by line number

remove ‘continental’ polymorphs …high backscatter electron intensity region… Cite reference for CITZAF

104-105. This is a pretty small difference (0.38 vs. 0.47); both indicate a good fit

…not a product… At first glance, this suggestions riesite forms upon a …. …gives the following… whereas? Can the authors provide a short description of what 17-20 GPa is based on? Provide reference for isobaric core condition

187, 189. riesite is spelled incorrectly

Two typos: -fluor…, and crstal Typo: -fluor…

Author Response

Review #1:

 Overview

This manuscript reports the discovery and documentation of a new TiO2 polymorph, Riesite. The mineral is an approved mineral by IMA, and is so named after the sample it was found in, which originated from the Ries impact structure in Germany. The authors present different types of diffraction data (X-ray microdiffraction, Rietveld refinement, EBSD), compositional data (EMPA), and some images (SEM-BSE) to document the crystallography of the discovered phase. In particular, emphasis in the Discussion is place on why this phase is not a variant of either srilankite or akaogiite.

In general, I found the manuscript overall well-prepared, the data appear to be of high quality, and the conclusions soundly argued. Knowledge of this phase will be of interest to planetary scientists, those that study impact craters, metamorphic petrologists, and other communities.

The main issues I identified in this manuscript that could benefit from attention are relatively minor, but that I think would help the presentation. These include some minor text editing, removing some typos, and a bit more sample-related documentation. In particular, I would like to see the addition of a few figures, so long as their inclusion does not result in the manuscript exceeding the length limit set by the journal. All things considered, my suggestion is to publish after moderate revision.

Response: We thank the reviewer for the positive and encouraging comments and the helpful suggestions! Below our responses to each comment in sequence:

Main issues:

(1) Can the authors include a TiO2 phase diagram with a suggestion for where riesite might reside? It is indicated in the text, but a phase diagram (even somewhat schematic) would be helpful.

Response: We agree with the reviewer that a reliable or even tentative phase diagram of titania in the 1-50 GPa pressure range would be a helpful addition. However, the phase boundary between akaogiite and srilankite is still not well constrained at high temperature, while the numerous compression studies at 300 K may in part reflect kinetic inhibition of transformation. This is nicely documented in the study by Al Khatatbeh et al. Phys. Rev. B, DOI: 10.1103/PhysRevB.79.134114, who even propose a completely different sequence of phase transformations (at odds with the large volume press studies cited in our paper). We feel, we cannot really contribute to this debate at this point nor should we draw a tentative diagram by giving some reference preference over others. If our interpretation of riesite as metastable backtransformation product of akaogiite is correct, there is no defined location of riesite within the TiO2 phase diagram.

For clarification we modified the Conclusion and write:

‘The sublattice disorder and the structural relation between akaogiite and riesite suggest that riesite is the product of retrograde transformation of akaogiite upon rapid decompression and cooling. In this case, riesite does not have a thermodynamic stability field but forms at elevated temperature under similar pressures as srilankite in dependence of the path (rapid decompression of akaogiite at elevated temperature). In fact, the riesite type material has been found in exactly this environment: a shock melt vein in an impact-metamorphosed bedrock fragment. The density of riesite is 4.37 ± 0.11 g/cm3, which is in between rutile (4.25 g/cm3) and srilankite (4.38 g/cm3), but below that of akaogiite at reference conditions (4.72 g/cm3).’

(2) The authors claim that reidite occurs adjacent to the riesite, and attribute petrographic description of the sample to Stahle et al. However, these authors do not mention reidite.

Response: We correct this statement. It was Erickson et al. (Microstructural constraints on the mechanisms of the transformation to reidite in naturally shocked zircon

By: Erickson, Timmons M.; Pearce, Mark A.; Reddy, Steven M.; et al. Contr. Min. Petr. 172,  Article Number: 6 (2017), who reported reidite from the Ries impact structure for the first time. We added this reference. There is a second report about reidite in a conference proceeding by Gucsik.

 Reidite is an important phase that interests a lot of people- can the authors include an image of a riedite-bearing zircon that occurs in the same sample? It adds another element to constrain the P history,

Response: We agree and add following statement to Materials&Methods:

Akaogiite [8] and reidite [31] had previously been reported from the Ries.

We also added following statement to the Discussion:

The presence of reidite (Fig 1d) is also consistent with pressures above 12 GPa, if temperatures were above 1200 K. Lower temperature shifts the transition from zircon to reidite to higher pressure [31]. We note that the shock-driven direct conversion along the Hugoniot of zircon and on experimental time scales of less than one ms occurs above 60 GPa [32].

 and does not appear to be documented anywhere else for this sample.

Response: Reidite was reported from the Ries but not from this xenolith. Since this xenolith ZLn114 is otherwise well documented, we agree that a figure and discussion about reidite is worth adding. We modified the paper accordingly (see above).

(3) The authors describe how riesite occurs together with ilmenite, rutile, and akaogiite- this sounds wildly interesting. Can the authors include an EBSD phase map to show this?

Response: We examined two riesite clasts as potential type materials for this new mineral. One of the clasts contains riesite and residual rutile. The other one riesite, akaogiite and ilmenite. The XRD patterns of rutile indicate high strain. However, the rutile kernel and also ilmenite are not exposed to the section surface. X-ray microdiffraction at 30 keV energy and in transmission mode probes the entire thickness of the thin section, whereas SEM/EBSD only penetrates some few 10s of nm. Akaogiite disorders nearly instantaneously under the e-beam during EBSD data collection.

 (4) The X-ray mapping is described as occurring over a 2x3 um2 area, that moved in 2 um steps. If the grain size of riesite is sub-micron, and also occurs as rims around ‘rutile kernals’, how was this dealt with given the much larger size of the X-ray beam? The authors should mention why this was/was not an issue in data collection/interpretation.

Response: The beamsize is optimal for examining polycrystalline phases of sub-micron grainsize. Riesite forms compact aggregates of submicron grains. Thus, the resulting diffraction patterns exhibit smooth Debye fringes whose integrated intensities are representative for the crystal structure of the material. This is what is needed for Rietveld refinement. If we cut the beam to ¼ micron^2, we obtain an array of reflections from hundreds of crystallites which evades any thorough structural analysis because of extinction by reflections of different crystallites and a rather poor reflection statistics for each individual crystallite. Also, centering on submicron crystallites is rather difficult for thinsections of rocks. Our approach provides the optimal statistics of structure-relevant information for this type of specimens.

(5) Many abbreviations occur in the manuscript. Some are followed by references, but many are not, and I think should be defined. Among these are EBSD, SEM, Rf, rMC, Rwp, Rp, and others.

Response: We change this and explain the abbreviations upon first occurrence.

(6) In Table 2, can the authors please include data for srilankite as well, to facilitate comparison (it would also be nice to include data for akaogiite).

Response: We agree that a direct comparison is useful. We add the structural parameters of srilankite (TiO2 endmember) to the table 2 caption. If we add it to the table, it may rather confuse readers – table 2 summarizes the results of the structure analysis of riesite but we did not analyse srilankite in this sample and this paper.

(7) In Figure 1, can the surrounding phases be identified? In panels A and B, can an inset box be included to show the locations of panels B and C? Can the authors include an EBSD map (band contrast, and/or phase map) of the grain shown in panel B? Its large enough that it should produce a lovely map, and would be the first of its kind. It would also inform about its state of strain.

An EBSD map of type riesite was not collected. It would be possible, but not within the time frame of revision provided by the journal.

(8) In the caption to Figure 2, switch the order of the first 4 words, as the first panels show X-ray images, and the second shows ebsd.

Response: Yes, we rearranged the Figure caption!

Specific comments by line number

remove ‘continental’

Response: Done!

polymorphs

Response: Corrected!

…high backscatter electron intensity region…

Response: Corrected!

Cite reference for CITZAF

Response: The reference was added.

104-105. This is a pretty small difference (0.38 vs. 0.47); both indicate a good fit

Response:  Yes, indeed. EBSD is unfortunately not a good tool for discriminating riesite and srilankite. Notably, the principal difference is in the site symmetry and factional atomic coordinates of Ti, which EBSD does not directly probe. We suspect that further occurrences of riesite pass unnoticed because EBSD is a common, available tool of analysis, whereas not every sample can be analysed at the synchrotron.

We added for clarification:

‘The main difference between both structures is in the site symmetry and fractional atomic coordinates of Ti. Since EBSD does not directly probe these parameters, it is not very sensitive to their differences.’

…not a product…

Response: Corrected!

At first glance, this suggestions riesite forms upon a ….

Response: Corrected!

…gives the following…

Response: Corrected!

whereas?

Response: In riesite the Ti sublattice is shifted to different stacking, compared to a fluorite-type disordered aristotype but the mechanism is described with fluorite as starting point not as opposite.

Can the authors provide a short description of what 17-20 GPa is based on?

Response: We are grateful for this question. In fact, Stahle et al. 2011 and 2017 only discuss shock pressures for amphibolite xenoliths but not for the restite(s). Hence, present is the first estimate of shock pressures. We modified the text for clarification:

‘In an earlier study of similar shock-melt veins in amphibolite xenoliths from Ries suevite Staehle et al. [22] estimated peak shock pressure up to 17-20 GPa based on garnet barometry but present is the first assessment of shock pressures for a xenolith of garnet-cordierite-sillimanite restite.’

Provide reference for isobaric core condition

Response: we added a reference.

187, 189. riesite is spelled incorrectly

Response: Corrected!

Two typos: -fluor…, and crstal Typo: -fluor…

Response: Both corrected!

Reviewer 2 Report

This work elucidated the structure of a new high-pressure polymorph of TiO2 and its phase relation among TiO2 high-pressure polymorph was discussed. This manuscript could be accepted after minor revision. Some comment which should be considered are given as follows.

At line 34, reference number [15, 15] should be corrected as [15, 16].

At line 67, the author wrote that “Quantitative elemental microanalyses (10) of these clasts were carried out at ~~~”. What does (10) mean?

In page 3 and 4, The notation of the space group symbols should be aligned. P1 2/c 1, P2/c and P12/c1 should be describe by the same notation.

At line 103, ‘F’ symbol at RF should be made subscript.

At line 133, Figure 4b should be corrected as Figure 2b.

In table 1, The definition of SFO and Biso were explained in caption.

In Figure caption, English captions need to be revised.

Is the position of “(b) Rieste clast.” correct?

Does Rieste mean Riesite?

The caption about (c) must be described.

In Figure 1, scale bars should be embedded into SEM images and highlighted.

At Figure 1(c), the author should increase he resolution of this photo if possible.

In Figure 2, (a), (b), (c) and (d) must be described in or near each figure.

Legends are required for all XRD patterns.

In line 198, “Rietveld refined model pattern or riesite,” should be “Rietveld refined model pattern of riesite.”

Please calculate the theoretical density of riesite and the another high-pressure polymorph of TiO2 from its lattice parameters and compare them. The author noted that riesite was metastable low-pressure phase of akaogiite. It is better to consider the phase-relation of TiO2 polymorphs from the viewpoint of density.

Author Response

Review #2:

Comments and Suggestions for Authors.

This work elucidated the structure of a new high-pressure polymorph of TiO2 and its phase relation among TiO2 high-pressure polymorph was discussed. This manuscript could be accepted after minor revision. Some comment which should be considered are given as follows.

 Response; We thank the reviewer for the positive and helpful comments and the thorough reading! Below, we address all comments and suggestions in sequence

At line 34, reference number [15, 15] should be corrected as [15, 16].

 Response: Corrected!

At line 67, the author wrote that “Quantitative elemental microanalyses (10) of these clasts were carried out at ~~~”. What does (10) mean?

 Response: We changed wording for clarification to

“Quantitative elemental microanalyses of these clasts were carried out at Caltech using a JEOL 8200 electron microprobe operated at 15 kV and 20 nA in focused beam mode. Analyses were processed with the CITZAF correction procedure. Ten point analyses of type riesite were averaged and the results are given in Table 1. ”.

Also, in the caption of table 1  we add:

“Analytical data for riesite. Average of ten point analyses. The empirical formula”

In page 3 and 4, The notation of the space group symbols should be aligned. P1 2/c 1, P2/c and P12/c1 should be describe by the same notation.

 Response: We are grateful for this comment and corrected this for clarification. First, we set the font of all space group symbols to italics, then we use short notation  where we talk about the space group only, long notation (P 1 2/c 1) where the setting of the space group is important – for instance in the group-subgroup mappings. We also fixed the notation of the space group of akaogiite by writing the translation component of  the screw axis as subscript.

At line 103, ‘F’ symbol at RF should be made subscript.

  Response: Corrected!

At line 133, Figure 4b should be corrected as Figure 2b.

Response: Corrected!

In table 1, The definition of SFO and Biso were explained in caption.

 Response: In the caption of table 1 we added the statement “. Site fractional occupancies (SFO) and isothermal displacement factors Biso are given”

In Figure caption, English captions need to be revised.

Response: We modified the Figure captions for clarification.

Is the position of “(b) Rieste clast.” correct?

Response: Corrected! We modified the caption text for clarification.

Does Rieste mean Riesite?

Response: Yes - corrected!

The caption about (c) must be described.

Response: Corrected (caption was there but mislabeled as “b”.

In Figure 1, scale bars should be embedded into SEM images and highlighted.

Response: We added the scale bars into the SEM images.

At Figure 1(c), the author should increase he resolution of this photo if possible.

 Response: This is not possible – the resolution is given by the instrument.

In Figure 2, (a), (b), (c) and (d) must be described in or near each figure.

Response: We moved the figures from the end of the paper to the discussion section and address each subfigure in the text. We also added missing subfigure labels a – d.

 Legends are required for all XRD patterns.

Response: We modified the figure caption to have all shown features explained.

In line 198, “Rietveld refined model pattern or riesite,” should be “Rietveld refined model pattern of riesite.”

Response: Meant” Rietveld refined pattern of riesite” ? We changed this.

Please calculate the theoretical density of riesite and the another high-pressure polymorph of TiO2 from its lattice parameters and compare them. The author noted that riesite was metastable low-pressure phase of akaogiite. It is better to consider the phase-relation of TiO2 polymorphs from the viewpoint of density.

Response: We agree. We added the density and a comparison to other TiO2 polymorphs:

‘The density of riesite is 4.37±0.11 g/cm3, which is in between rutile (4.25 g/cm3) and srilankite (4.38 g/cm3) but below that of akaogiite at reference conditions (4.72 g/cm3).’